



# An Examination of Enhanced Atmospheric Methane Detection Methods for Predicting Performance of a Novel Multiband Uncooled Radiometer Imager

Cody M. Webber[1] and John P. Kerekes[1]

[1]Digital Imaging and Remote Sensing Laboratory, Rochester Institute of Technology, Rochester, NY 14623, USA

**Correspondence:** Cody M. Webber (cmw3698@rit.edu)

**Abstract.** To evaluate the potential for a new uncooled infrared radiometer imager to detect enhanced atmospheric levels of methane, three different analysis methods were examined. A single pixel brightness temperature to NEdT comparison study performed using data simulated from MODTRAN6 revealed a single thermal band centered on the 7.68 um methane feature is capable of detecting the in band temperature contrast between a plume of about 17 ppm at ambient temperature and background
levels of methane at ambient temperature. Application of a normalized differential methane index method, a novel approach for methane detection, demonstrated how a simple two band method can be utilized to detect a plume of methane that is 10 ppm above ambient and -10 K from ambient temperature with $80\%$ hit rate and $17\%$ false alarm rate. This method was capable of detecting methane with similar levels of success as the third method, a proven multichannel method, Matched Filter. The matched filter approach was performed with six spectral channels. Results from these examinations suggest that given a high
enough concentration and temperature contrast, a multispectral system with a single band allocated to a methane absorption feature can detect methane.

## 1 Introduction

With the increased risk of climate change the value of global environmental monitoring has become increasingly important. Methane ($CH_4$), which naturally exists as the most abundant organic gas in the atmosphere   (Cicerone, 1988), is a potent
greenhouse gas with a radiative forcing per molecule approximately 20 times greater than carbon dioxide ($CO_2$) (Ramaswamy, 2001);   (Solomon et al, 2007). While the concentration of methane is lower than that of $CO_2$, the world has seen a rise in methane emissions since 2007, primarily from anthropogenic sources. Methane also has a moderately short lifespan in the atmosphere (about ten years), which means that efforts to reduce anthropogenic emission of methane would aid in slowing human contribution to climate change in a relatively short amount of time. The benefit of curbing methane emissions makes
it desirable to monitor likely sources of methane in order to quantify and limit emission from human activity   (Saunois et al, 2016).

     Airborne and satellite mounted remote imaging systems provide researchers with the ability to rapidly survey large swaths of Earth's surface and the atmospheric columns above the surface. This feature of remote imaging makes it a useful tool for monitoring atmospheric gas content and sources of rogue emissions. In the shortwave infrared, the Airborne Visible/Infrared



Imaging Spectrometer AVIRIS and its successor AVIRIS-NG are high spatial resolution, high spectral resolution imagers that have demonstrated the ability to detect enhanced levels of atmospheric methane by observing strong methane absorption features present between 2.0 and 2.5um   (Thrope et al, 2016). Longwave, or thermal, infrared hyperspectral imagery have been used to identify and track the movement of gas plumes in cluttered urban environments   (Broadwater et al, 2008). The Hy-

perspectral Thermal Emission Spectrometer (HyTES) is a high spectral resolution imager that has proved capable of detecting rogue methane emission sources by utilizing a clutter matched filter approach   (Hulley et al, 2016). The clutter matched filter method, when applied to HyTES imagery, has been proved capable of detecting enhanced levels of methane gas from both cluttered urban environments, such as the La Brea tar pits in Los Angeles, California, and from managed rural scenes, such as oil fields in Kern County, California. HyTES has also been used to develop an algorithm that can predict methane concentration

from thermal imagery   (Kuai et al, 2016). The satellite mounted systems, TROPOspheric Measuring Instrument (TROPOMI) and Greenhouse gases Observing Satellite (GOSAT) are capable of measuring global atmospheric methane content using solar backscattering   (Hu et al, 2018). The Methane Remote Sensing Lidar Mission (MERLIN) Minisatellite is scheduled to launch in 2020, and will utilize a short wave infrared source to detect methane plumes   (Jacob et al, 2016).

Improvements in remote thermal imaging systems and the design of new systems necessitates the evaluation of methane

detection capabilities. DRS Technologies has constructed a Multi-Band Uncooled Radiometer Imager (MURI) for the National Aeronautics and Space Administration's Instrument Incubator Program (IIP). MURI is designed to collect images in the thermal infrared which will be applied to the study of land surface climatology, soil moisture content, ecosystem dynamics, hazard and volcano emission ($SO_2$) monitoring, and methane detection   (Ely et al, 2016). The goal of this project is to demonstrate the value of utilizing low cost microbolometers in earth observation imaging systems. DRS Technologies aims to show that

implementing methods applied in the construction of MURI will reduce the cost and development time for airborne and space based imagers while maintaining a satisfactory performance in the thermal region of the infrared. A primary advantage of this design is that by utilizing a low cost microbolometer focal plane array the system does not require the installation of a potentially heavy and expensive cooling system. Two designs have been compiled for the MURI: an airborne demonstration system and a satellite mounted system. The system utilizes a $17\mu m/pixel$ microbolometer FPA, an integration time of $14\mu s$

and uses optics with an effective focal length of $120mm$ and an $f_{number}$ of 1. The design utilizes 6 spectral channels which are detailed in Table 1, along with DRS predictions of Noise Equivalent delta Temperature (NEdT) or the minimum brightness temperature difference each band can detect for the airborne instrument. MURI's band one has been allocated to be centered on a methane feature located around 7.68 um   (Ely et al, 2016). The inspiration for this study was to determine if it is possible to detect enhanced levels of atmospheric methane in the thermal infrared using a multispectral instrument with a single band

allocated to methane absorption features. To accomplish this, three different types of detection schemes were examined in order to predict performance and provide evidence for which methods provide useful results when applied to multispectral data from an instrument like MURI.



**Table 1.** MURI Band Allocations and Predicted Noise Equivalent delta Temperature

| Band # | Center Wavelength | Band Width | Predicted NEdT |
|:------:|:-----------------:|:----------:|:--------------:|
| B1 | 7.68um | 0.10um | 0.256K |
| B2 | 8.55um | 0.35um | 0.076K |
| B3 | 9.07um | 0.36um | 0.078K |
| B4 | 10.05um | 0.54um | 0.059k |
| B5 | 10.90um | 0.59um | 0.061K |
| B6 | 12.05um | 1.01um | 0.036K |

## 2 Methane Detection Method Descriptions

In this section, three methods of methane detection used to determine detectable cases for the uncooled instrument are described.

### 2.1 Single Pixel NEdT Comparison

The first study presented here investigates the potential contrast for a single thermal infrared pixel centered on the methane absorption feature present at 7.68 um. Here, a narrow bandpass of 100 nm is considered. The goal of the study is to determine under what scenarios a single band allocated to methane detection is capable of detecting the temperature difference indicative of an enhanced level of atmospheric methane.

In order to accomplish this, sensor reaching radiances were calculated using radiative transfer models produced with MOD-

TRAN6. This modeling code provides the ability to define a background surface, surface temperature, and atmosphere to calculate the spectral radiance that reaches a single pixel at the system's height. Utilizing the local chemical plume model option in MODTRAN6, spectral radiances, $L_{spec}$ were calculated for a background case, or a case without enhanced levels of methane, and a plume present case, or a case with enhanced levels of methane   (Berk et al, 2016).

Effective radiances, or the amount of light energy that the system is responsive to, can be calculated from the spectral

radiance:

$$L_{eff} = \frac{\int_{\lambda_i}^{\lambda_j} L_{spec} R d\lambda}{\int_{\lambda_i}^{\lambda_j} R d\lambda} \tag{1}$$

where $R$ is the responsivity of the pixel and $\lambda_i$ and $\lambda_j$ are the wavelength limits   (Schott, 2007). Note that this effective radiance is normalized by the responsivity curve of the spectral channel of the instrument. From effective radiance the brightness or



effective temperature can be calculated, which is the temperature perceived from the imaging system in reference to a black body.

$$T_{brightness} = \frac{hc}{\lambda_{center} k_b \log \frac{hc}{L_{eff} \lambda_{center}^5}} \tag{2}$$

where $h$ is Planck's constant, $c$ is the speed of light, $k_b$ is the Stephan Boltzman constant, and $\lambda_{center}$ is the center wavelength of the band  (Schott, 2007). By calculating a brightness temperature for both the background and plume present case and then taking the difference of the resultant brightness temperatures, a brightness temperature difference was found. Comparing this brightness temperature difference to the Noise Equivalent delta Temperature (NEdT), or the minimum brightness temperature difference the system is capable of detecting, reveals if the system would be able to detect the increased concentration of methane utilizing only the methane band.

## 2.2 Methane Detection Utilizing a Matched Filter

In order to better assess the system's methane detection capabilities, an approach proven to work for hyperspectral imagery was considered. The study presented here utilizes a matched filter approach to assess MURI's capability of detecting enhanced levels of atmospheric methane. While this method has been proven capable of detecting methane using thermal infrared HyTES data, applying the matched filter here is to investigate the viability of this method with a system with considerably fewer spectral bands (6 compared to 256) and only a single band allocated to the thermal infrared methane absorption feature.

The objective of developing a matched filter is to create a weighting function that when applied to an $N$ pixel by $n$ spectral channel radiance matrix, $L$, the output is a new image where intensity correlates with the presence of the signal of inquiry. Application of the matched filter begins with the assumption that there exists a signal, $b$, that is linearly superimposed on a background of the image, which can be written as:

$$r = \alpha b + c \tag{3}$$

where $r$ is the sensor reaching radiance, $\alpha$ is the strength of the spectral signal, and $c$ is a combination of noise and background signal  (Funk et al, 2001);  (Hulley et al, 2016). A realistic model of $c$ considers the correlation between spectral channels, which can be described in terms of the covariance matrix K:

$$K = <c'c'^T> = \frac{1}{N} L'L'^T \tag{4}$$





where $L'$ is mean subtracted radiance over all the pixels from matrix L. Knowing, $K$, the covariance of the image, the optimal matched filter can be matched to both the desired signal $b$ and the background, or "clutter". This clutter matched filter is defined as:

$$q = \frac{K^{-1}b}{\sqrt{b^T K^{-1} b}} \qquad (5)$$

It should be noted that $q$ is normalized so that if the signal is not present in the original image, the resultant matched filter image will prove to have a variance of 1. By applying the matched filter $q$ to the $N$ by $n$ matrix of radiances $L$, the clutter matched filter image is created:

$$CMFI = q^T L \qquad (6)$$

     After computing the CMFI, a simple threshold is applied to determine if the signal is present   (Funk et al, 2001);   (Hulley
et al, 2016). For this study, the threshold was varied in order to produce a Receiver Operator Characteristic (ROC) curve to assess the effectiveness of the method, rather than the effectiveness of a single threshold.

     The study presented here provides a comparison between the 6 channel multispectral MURI instrument and the 256 channel hyperspectral HyTES instrument when applying the matched filter to simulated imagery containing enhanced levels of atmospheric methane.

**2.3    Methane Detection Using a Normalized Differential Methane Index**

The final study described here aims to determine if detection of enhanced atmospheric methane is possible using information from pair of thermal spectral channels. This method has seen use in vegetation based studies in the form of the normalized difference vegetation index, or NDVI   (Rouse et al, 1973). Here, a normalized difference methane index, or NDMI, is calculated using the following equation:

$$NDMI = \frac{SB2 - SB1}{SB2 + SB1} \qquad (7)$$

     Where SB2 and SB1 are the radiance values recorded by two different spectral bands from the instrument, one that includes a methane feature and one that does not. The result is an image of intensity values that can be compared to threshold to determine if methane is present. If the plume is absorbing more thermal energy than passes through and is emitted by it, higher values for NDMI indicate a stronger likelihood of enhanced methane. If the plume is emitting more thermal energy than it absorbs,
which is characteristic of hotter plumes, lower values of NDMI indicate a stronger likelihood of enhanced methane. All cases considered for this study included plumes that produced spectral absorption features and therefore higher NDMI values were indicative of a stronger likelihood of enhanced methane.





For this study, two different band combinations were chosen to be compared. SB1 for each combination was MURI band 1. Centered at 7.68um, this band contains the strongest methane absorption feature. Two different bands were chosen for SB2 for comparison, the first was MURI band 2. This band was chosen as this band contains less methane absorption features than band 1 and covers a spectral region that has comparatively higher transmission than band 1. The other band chosen for SB2

was MURI band 6. This band was chosen because methane has the weakest effect on this band. After calculating the NDMI, a threshold varying from the lowest value pixel to the highest value pixel of the NDMI image was used to create ROC curves, which inform on how well the NDMI is an indicator of enhanced methane presence.

## 3   Data Set Creation and Validation

This section discusses the process of creating data sets for the three studies above and the validation of the simulated data.

### 3.1   Single-pixel Simulation Validation

In order to create a realistic data set of sensor reaching radiances for these studies, a scenario in which a rogue emission source has been detected was chosen to model the simulated data after. Data from HyTES collections are fitting for this purpose as the system's 256 spectral channels roughly cover the same region in the thermal infrared as the MURI design and collects over the methane absorption feature at 7.68 um. The chosen HyTES collection is shown in Fig. 1 and was recorded over Kern River oil

fields on 5 February 2015  (Jet Prupulsion Laboratory, 2019). The data set provided by JPL includes a flagged image, shown in Fig. 2, that identifies pixels that a matched filter predicted contained enhanced methane concentrations. Figure 3 shows a typical on and off plume spectra, as well as a simulated recreation of the data using MODTRAN6. The model was able to recreate the the HyTES spectra with a RMSE of $0.25 \frac{W}{m^2 sr \mu}$ for the methane present case and $0.15 \frac{W}{m^2 sr \mu}$ for the background case. Recreation of this data provided insight into reasonable scenes where methane could be detected, and informed the model

used in this study.

### 3.2   Matched Filter and NDMI Data Set Creation

To evaluate the multi-band methods, a simulated MURI image was created using higher spectral resolution HyTES imagery. By applying the MURI spectral response to the HyTES data, a six channel image with MURI's spectral channels was created. It should be noted that HyTES data does not fully cover the bandpass of MURI's band 6. The synthetic MURI image was created

using a subset of HyTES images recorded on 8 July 2014, which can be seen in Fig. 4. The chosen subset was determined to contain no detected enhanced methane pixels  (Jet Prupulsion Laboratory, 2019).

The images created by applying MURI's spectral response initially had less noise than the predicted noise for MURI. The noise in this image is defined as:

$$N_{simulated} = \frac{N_{HyTES}}{\sqrt{(\# of HyTES Bands)}} \tag{8}$$



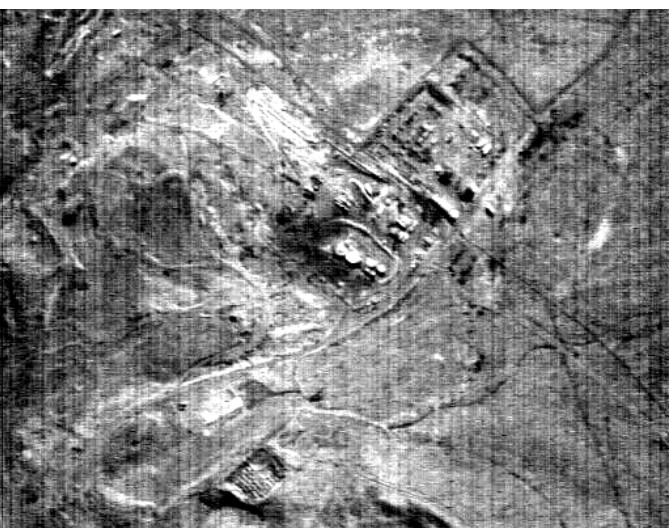

**Figure 1.** 7.68 HyTES band recorded on 5 February 2015. This image was used to validate our model method in MODTRAN 6

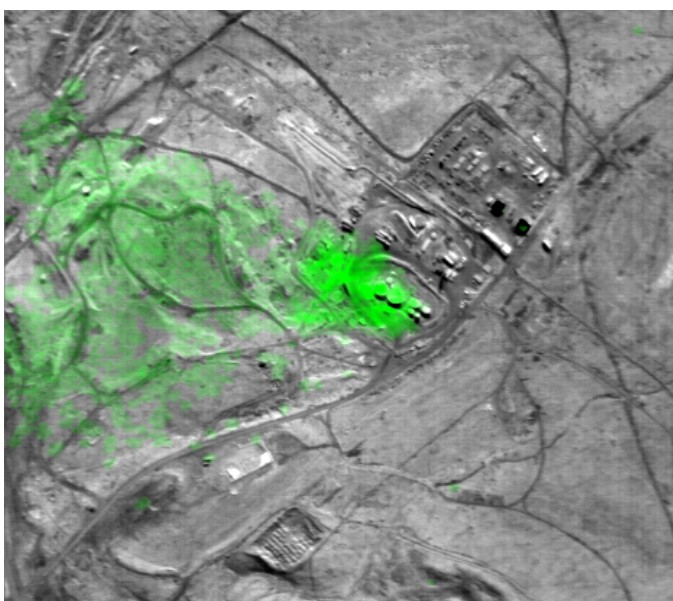

**Figure 2.** Flagged image, green indicates HyTES clutter matched filter has detected methane in that pixel.

This means that additional noise needed to be simulated in the image in order to better estimate a MURI image. The amount of additional noise can be defined as:

$$N_{add} = sqrt(N^2_{MURI} - (\frac{N_{HyTESimage}}{\sqrt{(\#ofHyTESBands)}})^2) \tag{9}$$



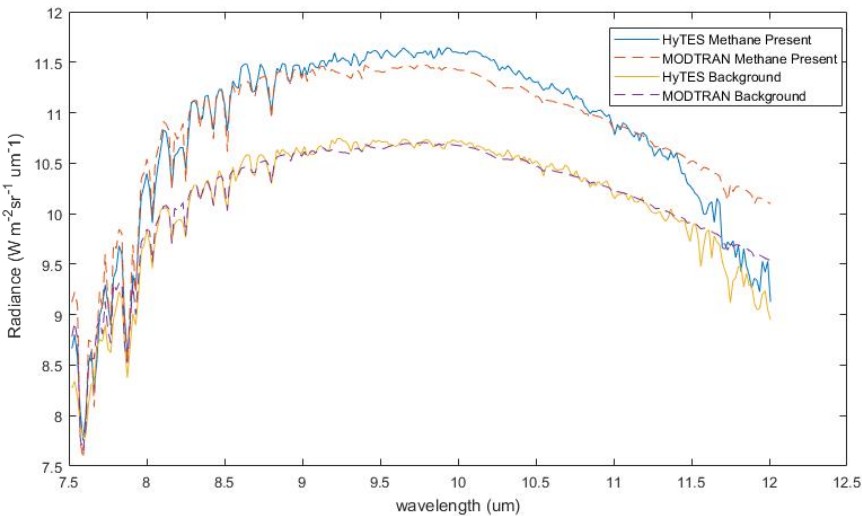

**Figure 3.** Recreation of HyTES spectra from data set used by Kuai et al. using MODTRAN 6. This is the recreation of one on plume and one off plume pixel from the 5 February 2015 data set.

This additional noise was calculated from the Noise Equivalent delta Temperature by first calculating Noise Equivalent delta Radiance:

$$NEdL = NEdT * \frac{dB}{dT} \tag{10}$$

Where $\frac{dB}{dT}$ is the derivative of Planck's Blackbody function with respect to temperature. The noise was then added to the image
by multiplying the difference in quadrature of the NEdLs with a Gaussian random number with mean 0 and standard deviation of 1.

The simulated dataset was created to determine the ability of MURI to detect higher concentration methane plumes. The data set for this investigation required an image with realistic variation and a known presence of methane. In order to accomplish this, a set of methane present HyTES images were created. These images were created using the local chemical plume model
of MODTRAN6, which outputs an on plume and off plume curve for at sensor radiance  (Berk et al, 2016). Both the off plume and on plume simulations were run with a limited atmosphere with only small amounts of $CO_2$. The off plume simulation contained only background levels of methane while the on plume model contained an enhanced concentration plume. Then a radiance difference was calculated between the off plume and on plume spectral curves, removing the effects of the small amount of $CO_2$ and background methane levels. The differences were then added to copies of the background HyTES image
to create a set of plume present images with realistic background variation and known quantity. The MURI images were then





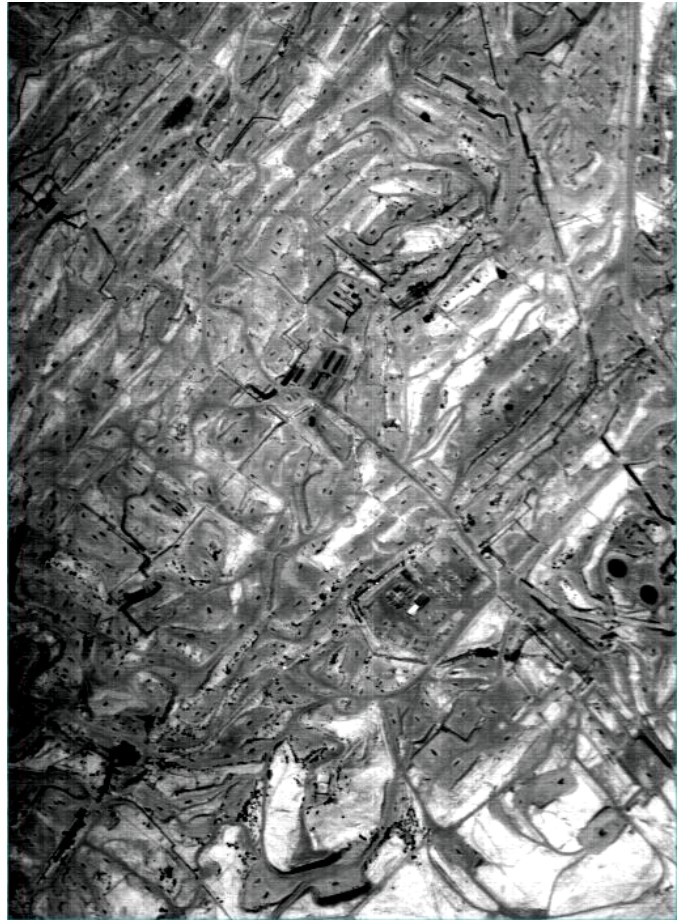

**Figure 4.** Single band view of HyTES image subset from 8 July 2014. This subset was used to produce the simulated dataset.

created by applying MURI's spectral response to the HyTES dataset. Additional noise was added to the images by the same method stated above.

# 4   Methane Detection Results

This section presents the results of applying each of the three methods for methane detection described above.

## 4.1   Single Pixel NEdT Study Results

For this study, a low altitude plume is considered. Spectral radiances in the methane band were simulated using MODTRAN6, as described in section 3. Table 3 contains a list of notable constants and their values which were derived from examining HyTES images, metadata, and the conditions under which the images were recorded    (Jet Prupulsion Laboratory, 2019).





**Table 2.** MODTRAN Simulation Chosen Values

| Constant | Value |
|---|---|
| Atmosphere | Midlatitude Summer |
| Water Vapor Scaling Factor | 0.10 |
| Collection Height | 4.572km (15000 feet) |
| Emitting Surface Temperature | 328 K |
| Plume Thickness | 20 m |
| Surface Emissivity | LAMB_SANDY_LOAM |
| Plume Height | 10 m |

Modern estimates of ambient atmospheric methane concentration are at about 1.8 ppm, dangerous levels for 8 hours of daily exposure to methane for humans is 1000 ppm, while the lower explosive limit is around 50,000 ppm. On the extreme end, methane levels of 500,000 ppm are noted to be the level in which humans experience asphyxiation. For this experiment the methane concentration within the plume was varied from 1 ppm to 50 ppm or $5\%$ the dangerous level for 8 hours of daily

5  exposure. The plume temperature was defined by a temperature difference to ambient temperature. The plume for this study varied from - 10 K to +10 K of ambient temperature at the plume height. Brightness temperature differences between the plume present and background cases were calculated. These differences were compared to the NEdT computed by DRS, which is 0.256 K for band one, or the methane feature band. The results of the low altitude plume model can be seen in Fig. 5.

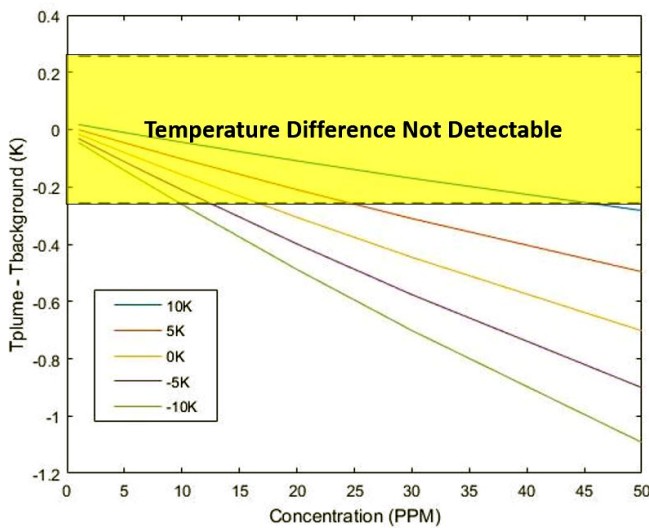

**Figure 5.** Low altitude plume model results displaying brightness temperature difference as a function of plume concentration. Figure identifies detectable and undetectable scenarios for MURI's predicted NEdT.





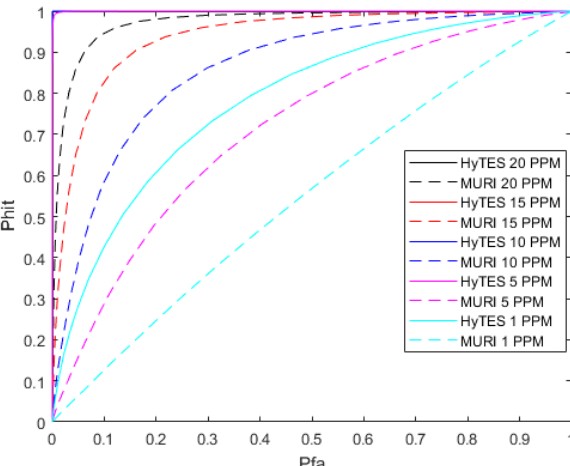

**Figure 6.** ROC curves for matched filter detection experiment. Results indicate high detection for most HyTES cases. Performance for MURI is high for 15 and 20 ppm, but low for 1 to 5 ppm.

The results here indicate that a plume with a temperature difference as high as +10 K to ambient temperature is absorbing energy. The higher temperature plumes require higher concentrations to detect, with the hottest in this study requiring 50 ppm plume to provide a detectable contrast (about 28 times background levels). At ambient temperature, a plume of about 17 ppm (about 10 times background levels) is required for the temperature difference to have a detectable contrast, and for the coldest plume temperature a concentration of 10 ppm creates a high enough temperature difference to display a detectable contrast. This study gives a baseline for detection for a single band allocated to methane absorption features.

### 4.2 Detection Using Matched Filter

For the purposes of this study, a low temperature plume (-10 K to ambient atmospheric temperature) with various concentrations of methane were simulated in the column and added to the background image containing only background levels of methane. The signal, b in equation 5, was defined as an absorbing methane plume and was extracted from the HITRAN dataset. The ROC curves in Fig. 6 provide an understanding of how well the system categorizes methane present pixels and background clutter using a the matched filter approach. The probability of false alarm ($P_{fa}$) indicates the fraction of background clutter pixels incorrectly categorized as methane present pixels, while the hit probability ($P_{hit}$) indicates the fraction of pixels correctly identified as methane present pixels. In an ideal circumstance, there exists a threshold value where $P_{hit}$ is 1 and $P_{fa}$ is zero.

Utilizing the matched filter approach shows HyTES is capable of detecting as low as 5ppm with a very false alarm rate. The hyperspectral system is even capable of detecting an additional plume of 1 ppm above background levels with a hit rate of $70\%$ and a false alarm rate of $30\%$. MURI's matched filter approach shows that an additional plume of 10 ppm can be detected with





80% accuracy and about 23% false alarm rate. Utilizing the matched filter on the two systems reveals that the narrow band hyperspectral system is outperforming the broader band multispectral system.

### 4.3 NDMI Detection Results

The result of applying the normalized differential methane index method for MURI data simulated from HyTES imagery and
MODTRAN6 can be seen in Fig. 7 and 8.

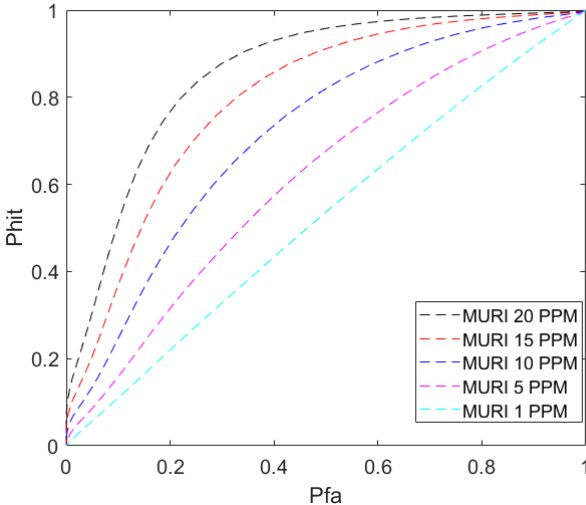

**Figure 7.** ROC curve describing the performance of applying the NDMI to the simulated data set using the methane feature band, band 1, and a relatively more transparent band, band 2.

    The results indicate the MURI system performs better using the NDMI using bands 1 and 6 than bands 1 and 2. The NDMI method performed on MURI band 1 and 6 performs as well as the matched filter approach being applied to all MURI bands, as the NDMI method shows 80% accuracy and about 17% false alarm rate for a scenario with an addition methane plume of 10ppm. This provides evidence indicating that given a high enough concentration and temperature contrast, a simple two band
approach can be used to detect enhanced levels of atmospheric methane with similar accuracy to a six band approach.

### 5   Conclusions

The studies detailed here predict the ability of an uncooled microbolometer imager to detect enhanced levels of atmospheric methane. The single band investigation confirmed that methane plumes with large concentrations and temperature differences compared to ambient atmospheric conditions lead to detectable contrasts, indicating that detection with a single pixel is possi-
ble, given the proper conditions. If a methane plume was large enough to be captured by multiple pixels, detection of plumes



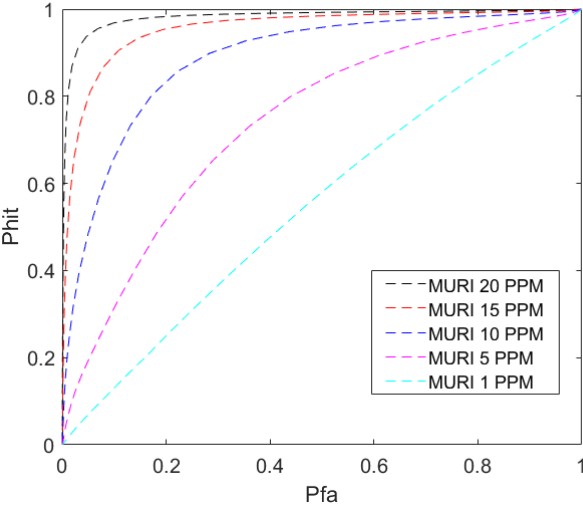

**Figure 8.** ROC curve describing the performance of applying the NDMI to the simulated data set using the methane feature band, band 1, and the MURI band with the least powerful methane signature, band 6.

with smaller temperature differences and methane concentrations could be possible by averaging over the pixels that collect plume signals. Future work includes examining additional scenarios, including different surface types and atmospheric parameters as well as validation of these results with the MURI system.

Application of the matched filter indicated the higher spectral resolution HyTES system would outperform the multispectral
5  MURI instrument. This study also shows that the NDMI approach provides similar detection using the multispectral MURI system. Given a significant quantity and temperature differential of methane, the NDMI performs well enough to be useful for a thermal imager with a single channel allocated for methane detection and a second band in a region with little overlap with a methane absorption feature. The results also indicate that the NDMI should be defined using one band that records in a region with a methane absorption feature and a broad channel that records in a region with no methane specific spectral features.
10  Future investigations aim to validate the results of these studies with images collected from test flights of the MURI system.

*Data availability.* The HyTES data used in this study can be requested from http://hytes.jpl. nasa.gov/order.

*Author contributions.* CW and JK designed the study. Modeling and experimental work was performed by CW under the supervision of JK. This paper was prepared by CW with assistance from JK.



*Acknowledgements.* The research described here was carried out at the Rochester Institute of Technology and was supported in part by the Earth Science Technology Office of the National Aeronautics and Space Administration under grant number 80NSSC18K0114. Any opinions, conclusions, recommendations, or findings described in this paper are those of the authors and do not reflect the views of the National Aeronautics and Space Administration.

5   *Competing interests.* The authors declare that they have no conflicts of interest.





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
