# Peer review of "An Examination of Enhanced Atmospheric Methane Detection Methods for Predicting Performance of a Novel Multiband Uncooled Radiometer Imager"

_Atmospheric Measurement Techniques, 2020_

## Referee Comment (RC1) · Anonymous Referee #1 · 29 May 2020

General comments: The manuscript uses three methods to evaluate the performance of a multiband uncooled radiometer imager, which would be cost-effective compared to a cooled hyperspectral instrument. The reasoning is sound, with interesting results for the community. There are however several clarifications that needs to be made regarding the aim of the study and for the evaluation of the instrument using the different methods to be more clear for the reader.

Specific comments: 1a. Is the goal of the study to be able to quantify CH4 concentrations (column densities) or to only detect areas of enhanced CH4 without quantifica-

tion? This should be made more clear. E.g. comparisons are made between MURI and HyTES, applying similar methods, but some studies using HyTES data have been able to quantify CH4 average concentrations using radiative transfer modeling (e.g. Kuai et al. 2016). If quantificaion is the aim, then e.g. for the Single Pixel NEdT Comparison: how would the method be able to differenciate between a change in temperature contrast and an actual increase in methane for the two cases (background and plume case). The brightness temperature would be affected by the background temperature, the CH4 temperature, and the CH4 column density. How can these three parameters be found from one brightness temperature? It is also likely that the background temperature (and thus the background vs CH4 temperature contrast) would be different between the two cases (no plume and plume).

1b. One of the aims of the study seems to be providing a novel, cost-efficient system for satellites, utilizing low cost microbolometers and not requiring an expensive cooling system. Is the developed system (MURI) mainly to be used for satellites (which are expensive anyway with many other high costs) or is the idea to use the system for ground-based (possibly airborne) measurements as well? (where reducing the cost could have a higher impact). This should be made more clear.

2. Section 2.3 (Normalized Differential Methane Index). The method does not account for varying ground emissivity, the background vs gas temperature contrast, or the H2O and N2O column densities (are there strong absorption lines from these in the the SB1 and SB2 bands?). These would effect the efficiency to detect CH4 (and for sure retrieval of column densities if this is a goal). If only detection, there could be false alarms from e.g. high H2O concentrations (which has lines overlapping the 7.7 micron CH4 band). Emissivities could also be different between different background materials.

3a. Section 4.1. There are three relevant temperatures for this test: background, ambient, and plume. The efficacy to detect a plume would be very dependent on the background - plume temperature contrast, and if this contrast is 0 deg the plume could not be detected regardless of the sensitivity of a sensor as all the CH4 absorption lines

in the plume emits as much light as is removed (resulting in no absorption and no detectable difference in brightness temperature). In Figure 5, the 0 K curve (plume - ambient contrast is 0) would be a horizontal line if Tambient = Tbackground. It should thus be made more clear what temperarure difference has been assumed between the ambient and background. Also, caption to Fig. 5 should explain that the curves are different contrast of ambient and plume temperature. In winter it could very well be emissions features, with the plume increasing the brightness temperature with e.g. a background of ice on a lake. The sentence (P11, first row) "The results here indicate that a plume with a temperature difference as high as +10 K to ambient temperature is absorbing energy" - this again depends on the ambient - background temperature (which is not given as the ambient temperature is not given). This should be made more clear.

3b. Similarly, in the conclusions (P12) it is stated "The single band investigation confirmed that methane plumes with large concentrations and temperature differences compared to ambient atmospheric conditions lead to detectable contrasts". I agree with sufficiently large concentrations, but the important temperature difference is not the ambient and plume temperatures, it is the background and plume temperatures. This should be made more clear/rewritten. One could easily have the case of a very large plume-ambient temperature difference (say 10 K) but also a 10 K background-ambient temperature difference, leading to no absorption lines and no difference in brightness temperature.

Technical corrections: - Wrong table number. Page 9, 4.1. states "Table 3 contains...", this should be Table 2? (there is no Table 3) - Abstract: 7.68 um -> 7.68 $\mu$m. Also in other parts of the text (e.g. Table 1) using u instead of $\mu$ - Table 1: Write $\mu$m in the headers instead of every row - Introduction: Use CH4 instead of methane after first having introduced "methane (CH4)". This is also the case for many other parts of the text. - P5L17. "from pair" - > "from pairs"

---

## Referee Comment (RC2) · Anonymous Referee #2 · 8 Jun 2020

**1  General Comments**

The manuscript (Webber and Kerekes 2020) compares the performance of three different analytical methods for detecting methane in remote sensing imagery taken using an uncooled multispectral infrared (IR) radiometer. Given the prohibitive cryogenic requirements of traditional thermal IR imagers, an uncooled instrument would lower barriers to deploying imagers for atmospheric methane detection. This paper provides a useful evaluation of this system for methane detection; however, the paper has gaps in

the description of the methodology, and the discussion and conclusions require more development. In particular, more quantitative details about assumptions made and model input used should be included, and reasons for the values chosen should be explained.

**2 Specific Comments**

**Page 2**

l. 9-10: The phrasing that HyTES has been used to develop an algorithm that can predict methane concentration from thermal imagery is somewhat vague and therefore confusing. It would be more helpful to identify the improvements in the HyTES retrieval algorithm in Kuai et al. (2016) that are most relevant to the research described in this paper.

l. 19: Given that sensors that operate in various regions of the IR spectrum are discussed, it would be helpful to briefly clarify why traditional thermal IR sensors require cooling and the advantage of thermal IR over shortwave infrared (SWIR) sensors, which also measure methane but do not have the same cooling requirements.

l. 21: What defines a "satisfactory performance"? What is the level of sensitivity, precision, accuracy, or another relevant metric needed for methane detection applications of MURI?

l. 23: What is the difference between the airborne and satellite system? Are they using the same FPA?

**Page 3**

l. 13: What assumptions were made about environmental conditions, particularly the concentrations of interfering molecules such as water vapor?

[Figure]

**Page 4**
l. 5-9: More details are needed for the methodology, particularly what assumptions were made in modelling the background and plume-present cases and why those assumptions were chosen. A discussion of the sensitivity of the model output to these assumptions should be included here if some a priori knowledge of the sensitivity factored into the choice of assumptions, and/or in the Results/Discussion section if relevant to determining the validity of the results.
l. 8: What is the magnitude of the increased concentration of methane? How does this compare to the Noise Equivalent Concentration Length (NECL) and/or minimum detectable column density of the sensor?
l. 13-15: Since only a single band is allocated to the methane feature, what is the purpose of the other bands? Section 2.3 demonstrates that the other bands can help constrain the methane retrieval, but if they have additional functions, those functions should be listed (in this paragraph, in the general description of the instrument, or in Table 1).
l. 21: Units associated with each of the variables would be helpful to conceptualize the relationships in Equation 3 and clarify what is meant by "signal", which can refer to multiple aspects of the data stream.

**Page 5**
l. 9-11: Is the threshold applied to the CMFI value, or some statistics associated with it, such as a confidence interval or t-stat? Also, please provide a short explanation of how the ROC curve is used to assess the effectiveness of the method.
l. 23-27: This explanation is somewhat confusing. Is this paragraph describing whether the methane feature is giving an absorption versus emission signal in the detection? The way that NDMI is described, it seems like it would be possible to have negative values that can be indicative of a methane plume, and if no plume exists, the NDMI would be zero. If so, it seems that a higher absolute value of the NDMI would indicate

higher methane. Please clarify.

**Page 6**

l. 4: Please specify what band 2 has a comparatively higher transmission of: the atmosphere, instrument filter, etc.

l. 17: It's unclear what is meant by "on and off plume spectra". Are these the spectra for a single background pixel and a different pixel that has a methane detection? Also, what are the assumptions that were made for the MODTRAN simulated recreation of the data? If these are the same assumptions used in Table 2, please refer to that table in this paragraph.

l. 18: Based on the radiance values in Figure 3, the RMSE for the methane plume case is about 2.5% – How does this contribute to the uncertainty in the methane column density amount?

l. 19: How is "reasonable" defined? High confidence? If so, what is the threshold?

**Page 8**

l. 11: Is "only small amounts of $CO_2$" referring to the ambient concentration input into MODTRAN? Please provide the actual value used and why it was chosen. If these results are not sensitive to the assumed concentration of $CO_2$, please state that; if the chosen concentration of $CO_2$ impacts the results, however, provide justification for the value chosen (e.g. regional average concentrations taken from in situ or satellite measurements).

l. 12: What is the level of enhancement in the "enhanced concentration plume"? Please be quantitative.

**Page 10**

Table 2: Change "Plume Height" to "Plume Altitude", as the former could be confused with "Plume Thickness". Also, please add a row with the assumed ambient temperature.

Section 4.1: Please provide rationale for the model inputs listed in Table 2 in this section. For instance, was the plume thickness derived from data, a model, or experiments? Was the ambient temperature measured locally, or taken from a model, and if so, which one? Assumptions that don't have a significant impact on results can be stated as such; however, justification should be provided for assumptions that alter the results and especially conclusions of this paper.

**Page 11**

l. 1-6: It is unclear why the sensitivity described was framed in terms of the temperature gradient between the plume and the ambient air, as it is the temperature gradient between the surface and the plume that drives the sensitivity in hyperspectral imagery. There might be a reason to frame the conclusions in the terms used, but it is difficult to evaluate those conclusions without knowing what ambient air temperature was chosen. The minimum detectable concentration of methane is lower when the thermal contrast between the plume and the surface is high; for example, a very hot plume should be more detectable over a low-temperature surface. Thus, the assertions made in this paragraph would not apply in all cases and would depend on the relative ambient, surface, and plume temperatures. Since the paper is evaluating the performance of new instrumentation, characterizing which conditions the conclusions hold for would be helpful in evaluating the applicability of these techniques for conditions that deviate from those chosen for this study.

l. 12-14: When determining the false positive rate, what is used as truth? Is there ground-truth, or is the HyTES detection mask considered truth? Also, what is the region of interest threshold chosen? That is, does the algorithm require a certain number of contiguous pixels with methane detection before the plume is accepted?

[Figure]

**3   Technical Suggestions**

**Page 1**
l. 16-17: A citation is missing after "While the concentration of methane is lower than that of CO2, the world has seen a rise in methane emissions since 2007, primarily from anthropogenic sources."

**Page 2**
l. 3: "Thrope" should be changed to "Thorpe".
l. 5: It would be useful to specify that HyTES is a longwave infrared (LWIR) imager.
l. 18: Since the abbreviation for methane, $CH_4$, is used earlier in the paper, it should be continue to be used consistently. This applies to the remainder of the paper, as well.
l. 11: A comma is missing after "(GOSAT)".
l. 17: A comma is missing after "infrared".
l. 24: "FPA" should be changed to "focal place array (FPA)".
l. 25: A comma is missing after "channels".
l. 28: Both $\mu m$ and um are used in this paragraph. One convention should be chosen for the entire paper.

**Page 3**
l. 5: Elsewhere in the paper, pixel is also used to describe both the physical pixel on the FPA (e.g. page 2) and the spatial pixel in the image (e.g. page 4). For clarity, change this instance and other references to the spectral pixel to "channel", which is used later in the paper.

**Page 4**
l. 16: Add "spatial" between "N" and "pixel".

**Page 5**

l. 21-22: Specify whether SB2 or SB1 includes the methane feature.

**Page 6**

l. 11-12: Specify what "after" is referring to in the sentence "[...] a scenario in which a rogue emission source has been detected was chosen to model the simulated data after."

l. 15: Change "Prupulsion" to "Propulsion" (applies to other instances of this citation in the manuscript).

**Page 7**

Figure 1: Specify units after "7.68". Also please add the ground sampling distance (GSD) of the image.

**Page 8**

l. 10: Remove the typo in "for at sensor radiance".

**Page 9**

l. 7: Change the reference to "Table 3" to "Table 2".

**Page 10**

l. 1-2: Citations are needed after "Modern estimates of ambient atmospheric methane concentration are at about 1.8 ppm, dangerous levels for 8 hours of daily exposure to methane for humans is 1000 ppm, while the lower explosive limit is around 50,000 ppm."

l. 8: Remove "or" before "the methane feature band".

**Page 11**

l. 15: An adjective is missing between "very" and "false".

**Page 12**

l. 14: Change "pixel" to "channel" or "band" if spectral pixel is what is meant.

---

## Author Comment (AC2) · 6 Jul 2020

Referee #2 Comments Response: On behalf of myself and my coauthor I would like to thank you for taking the time to review and provide feedback on our paper. Your feedback was very helpful and much appreciated. We know that your feedback will help make our revision a better discussion of our work.

General Comments The manuscript (Webber and Kerekes 2020) compares the performance of three different analytical methods for detecting methane in remote sensing

imagery taken using an uncooled multispectral infrared (IR) radiometer. Given the prohibitive cryogenic requirements of traditional thermal IR imagers, an uncooled instrument would lower barriers to deploying imagers for atmospheric methane detection. This paper provides a useful evaluation of this system for methane detection; however, the description of the methodology, and the discussion and conclusions require more development. In particular, more quantitative details about assumptions made and model input used should be included, and reasons for the values chosen should be explained

Author Response(AR): Thank you for your comments pointing out the potential benefits of an uncooled instrument and recognizing the utility of our studies. We have added details to the descriptions of the methodology and enhanced our discussions to clarify points raised by the reviewers.

Specific Comments Page 2 l. 9-10: The phrasing that HyTES has been used to develop an algorithm that can predict methane concentration from thermal imagery is somewhat vague and therefore confusing. It would be more helpful to identify the improvements in the HyTES retrieval algorithm in Kuai et al. (2016) that are most relevant to the research described in this paper

ARl: 9-10 Some of the data used to inform the models we used in this study is from Kuai et al. We reference this study here to give context of what thermal instruments have been used for in the past as attempting to retrieve methane concentration is out of the scope of this study.

l. 19: Given that sensors that operate in various regions of the IR spectrum are discussed, it would be helpful to briefly clarify why traditional thermal IR sensors require cooling and the advantage of thermal IR over shortwave infrared (SWIR) sensors, which also measure methane but do not have the same cooling requirements.

l:19 A brief discussion of the advantages of TIR over SWIR was added and a statement of why traditional thermal IR sensors require cooling was added to this section.

l. 21: What defines a "satisfactory performance"? What is the level of sensitivity, precision, accuracy, or another relevant metric needed for methane detection applications of MURI?

ARl:21 No specific quantitative metrics for environmental applications were defined for the MURI project. Rather, satisfactory performance was defined more qualitatively as the system demonstrating useful performance in environmental applications. The project was conceived primarily as a technology development effort. This has been clarified in our revision.

l. 23: What is the difference between the airborne and satellite system? Are they using the same FPA?

ARl:23 The airborne and the satellite system design utilize the same focal plane array and similar optics with an effective focal length of 120 mm and an fnumber of 1. The work presented here is focused on the airborne system and we have clarified that in our revision.

Page 3: l. 13: What assumptions were made about environmental conditions, particularly the concentrations of interfering molecules such as water vapor?

ARl:13 This has been clarified in the Data Set Creation section with environmental conditions specified in Tables 2, 3, and 4.

Page 4: l. 5-9: More details are needed for the methodology, particularly what assumptions were made in modelling the background and plume-present cases and why those assumptions were chosen. A discussion of the sensitivity of the model output to these assumptions should be included here if some a priori knowledge of the sensitivity factored into the choice of assumptions, and/or in the Results/Discussion section if relevant to determining the validity of the results.

ARl:5-9 Modeling of the background was done by choosing parameters which would match the model output to the HyTES scene. Additional details have been included in

section 3 to address the model parameters and their derivations.

l. 8: What is the magnitude of the increased concentration of methane? How does this compare to the Noise Equivalent Concentration Length (NECL) and/or minimum detectable column density of the sensor?

ARl: 8 For the single pixel study, NEdT is chosen as the metric of comparison for detection as we are changing plume temperature and concentration and NECL will change for plumes of different temperature. Additional methane plumes were varied from 1 to 50 ppm. The graph was created using the following concentration plumes: 1, 5, 10, 20, 30, 50 ppm. The step sizes were chosen to ensure fine sampling such that the shape of the Tplume – Tbackground curves were not affected by the concentration intervals.

l. 13-15: Since only a single band is allocated to the methane feature, what is the purpose of the other bands? Section 2.3 demonstrates that the other bands can help constrain the methane retrieval, but if they have additional functions, those functions should be listed (in this paragraph, in the general description of the instrument, or in Table 1).

ARl:13-15: Additional information for each band has been added to Table 1 in order to describe their functions.

l. 21: Units associated with each of the variables would be helpful to conceptualize the relationships in Equation 3 and clarify what is meant by "signal", which can refer to multiple aspects of the data stream

ARl: 21 The signal is defined here as the spectral signature of a methane plume, which here is defined as absorption intensity, retrieved from hitran dataset. A clarifying statement as to what the signal is has been added. This assumed linear relationship is written in terms of sensor reaching radiance, so r and c are in terms of W/m2sr um, b is a absorption intensity, and $\alpha$ is a scalar proportional to the plume strength.

Page 5: l. 9-11: Is the threshold applied to the CMFI value, or some statistics associated with it, such as a confidence interval or t-stat? Also, please provide a short explanation of how the ROC curve is used to assess the effectiveness of the method.

ARl: 9-11 The threshold is applied to the CMFI value. The ROC curves describes the hit and false alarm rate at each concentration, indicating where in the concentration space the system can reliably differentiate between on and off plume pixels. Another method of assessing ROC curves is to calculate the area under the curves, which we've provided in the supplementary materials. Section 4.2 includes a discussion of how the ROC curves are used to assess the effectiveness of the method, and the section has been updated in response to your suggestion.

l. 23-27: This explanation is somewhat confusing. Is this paragraph describing whether the methane feature is giving an absorption versus emission signal in the detection? The way that NDMI is described, it seems like it would be possible to have negative values that can be indicative of a methane plume, and if no plume exists, the NDMI would be zero. If so, it seems that a higher absolute value of the NDMI would indicate C3 higher methane. Please clarify.

ARl: 23-27 Thank you for this comment, we have added to the paragraph to clarify. This paragraph is meant to describe how the NDMI will be different for a plume that is hotter than the surface and a plume that is cooler than the surface. If a plume is hotter than the surface, the resulting at sensor radiance for the methane feature band (SB1) will be higher than if there was no plume present. This would mean that, given a scenario in which all other variables remain the same, NDMI would be a lower value as the difference between SB2 and SB1 will be lower and the sum of SB2 and SB1 will be higher. If the plume is cooler than the background surface temperature, the plume will absorb more energy than it emits and the at sensor radiance for the methane feature band (SB1) will be lower than if there was not a plume present. NDMI would then be a higher number as the difference between SB2 and SB1 will be higher and the sum of SB2 and SB1 will be lower than the plume not present case. The NDMI is only 0

if SB1 and SB2 are equal and is not necessarily an indication of a plume presence or absence as SB1 and SB2 are defined as different spectral channels. It is possible to have a negative NDMI given a high enough temperature difference between the plume and the background surface. The NDMI is a relative measurement, and therefore can only be determined by comparing NDMI calculations across an image.

Page 6 l. 4: Please specify what band 2 has a comparatively higher transmission of: the atmosphere, instrument filter, etc.

ARl: 4 We have clarified that we mean the atmosphere has a higher transmission for band 2 than band 1.

l. 17: It's unclear what is meant by "on and off plume spectra". Are these the spectra for a single background pixel and a different pixel that has a methane detection? Also, what are the assumptions that were made for the MODTRAN simulated recreation of the data? If these are the same assumptions used in Table 2, please refer to that table in this paragraph

ARl: 17 The on and off plume spectra describe two pixels taken from the HyTES imagery shown in Figure 1. The on plume pixel refers to a pixel identified by the HyTeS dataset to contain a methane plume, and the off plume pixel refers to a pixel identified by the HyTES data set to not include an enhanced level of atmospheric methane. Some assumptions are shared between this simulation and the original Table 2, a new table has been included to specify the chosen values.

l. 18: Based on the radiance values in Figure 3, the RMSE for the methane plume case is about 2.5% – How does this contribute to the uncertainty in the methane column density amount?

ARl: 18 Our single spectral band approach does not allow for quantification of methane, making column density uncertainties outside the scope of this study.

l. 19: How is "reasonable" defined? High confidence? If so, what is the threshold?

ARl:19 The language in our manuscript was unintentionally vague. We meant to say this recreation gave us insight into scenes where enhanced methane has been detected before and the confidence to utilize MODTRAN to create controllable simulations that resemble real data that has been used to detect methane previously. The language has been changed to better reflect the intended meaning.

Page 8 l. 11: Is "only small amounts of CO2" referring to the ambient concentration input into MODTRAN? Please provide the actual value used and why it was chosen. If these results are not sensitive to the assumed concentration of CO2 please state that; if the chosen concentration of CO2 impacts the results, however, provide justification for the value chosen (e.g. regional average concentrations taken from in situ or satellite measurements).

ARl:11 As stated on line 13-14 the effect of CO2 absorption is eliminated by our data set creation approach. The results are not sensitive to the assumed concentration of CO2 in this simulation. The chosen value is a small amount of CO2 that must be included to allow the MODTRAN simulation to run properly.

l. 12: What is the level of enhancement in the "enhanced concentration plume"? Please be quantitative.

ARl: 12 The enhanced concentration plumes refer to the varying quantities used in the experiments, which are identified in Figures 6, 7, and 8. The enhanced quantities vary from 1 – 20 ppm and this has been clarified in our revision.

Page 10 Table 2: Change "Plume Height" to "Plume Altitude", as the former could be confused with "Plume Thickness". Also, please add a row with the assumed ambient temperature

ARTable 2: Thank you, your suggestion has been added to Table 3 (formerly table 2).

Section 4.1: Please provide rationale for the model inputs listed in Table 2 in this section. For instance, was the plume thickness derived from data, a model, or experi-

ments? Was the ambient temperature measured locally, or taken from a model, and if so, which one? Assumptions that don't have a significant impact on results can be stated as such; however, justification should be provided for assumptions that alter the results and especially conclusions of this paper.

AR_Section 4.1: Model assumptions were derived by adjusting MODTRAN inputs to produce simulated radiances that matched the HyTES data. Ambient temperature was recorded by a local weather station and retrieved from Wunderground. Additional rationale has been included to inform their reason for inclusion in the study in our revision.

Page 11 l. 1-6: It is unclear why the sensitivity described was framed in terms of the temperature gradient between the plume and the ambient air, as it is the temperature gradient between the surface and the plume that drives the sensitivity in hyperspectral imagery. There might be a reason to frame the conclusions in the terms used, but it is difficult to evaluate those conclusions without knowing what ambient air temperature was chosen. The minimum detectable concentration of methane is lower when the thermal contrast between the plume and the surface is high; for example, a very hot plume should be more detectable over a low-temperature surface. Thus, the assertions made in this paragraph would not apply in all cases and would depend on the relative ambient, surface, and plume temperatures. Since the paper is evaluating the performance of new instrumentation, characterizing which conditions the conclusions hold for would be helpful in evaluating the applicability of these techniques for conditions that deviate from those chosen for this study.

ARl: 1-6 . Results were framed as plume to ambient atmospheric temperature differences as this is how our models are defined. We have included the ambient temperature in our revision and have describe our results with context of surface and plume temperature differential. Additionally, a supplemental figure has been added to display the results in terms of the plume/background temperature difference.

l. 12-14: When determining the false positive rate, what is used as truth? Is there ground-truth, or is the HyTES detection mask considered truth? Also, what is the region of interest threshold chosen? That is, does the algorithm require a certain number of contiguous pixels with methane detection before the plume is accepted?

ARI:12-14 As described in Section 3, MURI images were simulated using a HyTES image containing no enhanced concentration methane plumes. One image was created to be the background image, and a set of images were created simulating enhanced concentration methane plumes. We therefore have two separate images, one with methane and one without. This makes for an easy identification of hits, false alarms, correct rejections, and misses. Section 3 has been updated to clarify how the truth map is defined.

Technical Suggests: Page 1 l. 16-17: A citation is missing after "While the concentration of methane is lower than that of CO2, the world has seen a rise in methane emissions since 2007, primarily from anthropogenic sources." Page 2 l. 3: "Thrope" should be changed to "Thorpe". l. 5: It would be useful to specify that HyTES is a longwave infrared (LWIR) imager. l. 18: Since the abbreviation for methane, CH4, is used earlier in the paper, it should be continue to be used consistently. This applies to the remainder of the paper, as well. l. 11: A comma is missing after "(GOSAT)". l. 17: A comma is missing after "infrared". l. 24: "FPA" should be changed to "focal place array (FPA)". l. 25: A comma is missing after "channels". l. 28: Both $\mu$m and um are used in this paragraph. One convention should be chosen for the entire paper. Page 3 l. 5: Elsewhere in the paper, pixel is also used to describe both the physical pixel on the FPA (e.g. page 2) and the spatial pixel in the image (e.g. page 4). For clarity, change this instance and other references to the spectral pixel to "channel", which is used later in the paper. Page 4 l. 16: Add "spatial" between "N" and "pixel"

Page 5 l. 21-22: Specify whether SB2 or SB1 includes the methane feature. Page 6 l. 11-12: Specify what "after" is referring to in the sentence "[...] a scenario in which a rogue emission source has been detected was chosen to model the simulated data

after." l. 15: Change "Prupulsion" to "Propulsion" (applies to other instances of this citation in the manuscript). Page 7 Figure 1: Specify units after "7.68". Also please add the ground sampling distance (GSD) of the image. Page 8 l. 10: Remove the typo in "for at sensor radiance". Page 9 l. 7: Change the reference to "Table 3" to "Table 2". Page 10 l. 1-2: Citations are needed after "Modern estimates of ambient atmospheric methane concentration are at about 1.8 ppm, dangerous levels for 8 hours of daily exposure to methane for humans is 1000 ppm, while the lower explosive limit is around 50,000 ppm." l. 8: Remove "or" before "the methane feature band".

Page 11 l. 15: An adjective is missing between "very" and "false". Page 12 l. 14: Change "pixel" to "channel" or "band" if spectral pixel is what is meant.

AR Technical Suggestions: Thank you for the technical suggestions and for being so thorough. We are grateful and believe the additional technical suggestions you've provided have certainly improved the quality of the paper.

Please also note the supplement to this comment:
https://www.atmos-meas-tech-discuss.net/amt-2020-53/amt-2020-53-AC2-supplement.pdf

**Supplement:**

**Supplemental Materials**

Cody M Webber[1] and John P Kerekes[1]

[1]Digital Imaging and Remote Sensing Laboratory, Rochester Institute of Technology, Rochester, NY 14623, USA

**Correspondence:** Cody M Webber (cmw3698@rit.edu)

[Figure]

**SM- 1.** Recreation of Figure 5. Describes NEdT comparison study using temperature difference between the plume and the surface.

**ST- 1.** Area Under MURI Matched Filter ROC Curves (Figure 6)

| Concentration PPM # | Line Color | Area |
|---|---|---|
| 20 | Black | 0.97 |
| 15 | Red | 0.94 |
| 10 | Blue | 0.86 |
| 5 | Pink | 0.71 |
| 1 | Cyan | 0.55 |

**ST- 2.** Area Under MURI NDMI Band 1 and 2 ROC Curves (Figures 7)

| Concentration PPM # | Line Color | Area |
|---|---|---|
| 20 | Black | 0.86 |
| 15 | Red | 0.8 |
| 10 | Blue | 0.72 |
| 5 | Pink | 0.62 |
| 1 | Cyan | 0.52 |

**ST- 3.** Area Under MURI NDMI Band 1 and Band 6 ROC Curves (Figure 8)

| Concentration PPM # | Line Color | Area |
|---|---|---|
| 20 | Black | 0.98 |
| 15 | Red | 0.95 |
| 10 | Blue | 0.89 |
| 5 | Pink | 0.74 |
| 1 | Cyan | 0.55 |

---

## Author Response (AR1)

Referee #1 Comments Response:

On behalf of myself and my coauthor I would like to thank you for taking the time to review and provide feedback on our paper. Your feedback was very helpful and much appreciated. We know that your feedback will help make our revision a better discussion of our work.

General comments: The manuscript uses three methods to evaluate the performance of a multiband uncooled radiometer imager, which would be cost-effective compared to a cooled hyperspectral instrument. The reasoning is sound, with interesting results for the community. There are however several clarifications that needs to be made regarding the aim of the study and for the evaluation of the instrument using the different methods to be more clear for the reader.

AR General Comments: Thank you for your summary and we agree the results will be of interest to the community. We have made a number of changes in response to reviewer comments and the revised version is more clear.

**Specific Comments:**

1a. Is the goal of the study to be able to quantify CH4 concentration? This should be made more clear. E.g. comparisons are made between MURI and HyTES, applying similar methods, but some studies using HyTES data have been able to quantify CH4 average concentrations using radiative transfer modeling (e.g. Kuai et al. 2016). If quantification is the aim, then e.g. for the Single Pixel NEdT Comparison: how would the method be able to differentiate between a change in temperature contrast and an actual increase in methane for the two cases (background and plume case). The brightness temperature would be affected by the background temperature, the CH4 temperature, and the CH4 column density. How can these three parameters be found from one brightness temperature? It is also likely that the background temperature (and thus the background vs CH4 temperature contrast) would be different between the two cases (no plume and plume).

AR1a. The goal of this study is to detect enhanced levels of atmospheric CH4 without quantification. The purpose of the single pixel NEdT is to identify possible scenarios that lead to absolute brightness temperature differences higher than MURI's band one Noise Equivalent delta Temperature. We have changed some of the language in each section to clarify these goals.

Changes: Page 1, Lines 3- 6 previously read: "...single thermal band centered on the 7.68 um methane feature is capable of detecting the in band temperature contrast between a plume of about 17 ppm at ambient temperature and background levels of methane at ambient temperature" Now reads: "...single thermal band centered on the 7.68  $\mu$ m methane feature leads to a detectable brightness temperature difference exceeding the sensor noise level for a plume of about 17 ppm at ambient atmospheric temperature compared to an ambient plume with no enhanced methane present.

Page 12 line 11 previously read: "how well the system categorizes methane" Now reads: "how well the system distinguishes methane" Line 15: Refers to method as "detection scheme"

1b. One of the aims of the study seems to be providing a novel, cost-efficient system for satellites, utilizing low cost microbolometers and not requiring an expensive cooling system. Is the developed system (MURI) mainly to be used for satellites (which are expensive anyway with many other high costs)

or is the idea to use the system for ground-based (possibly airborne) measurements as well? (where reducing the cost could have a higher impact). This should be made more clear.

AR1b. The MURI system was designed to demonstrate the value of utilizing low cost microbolometers in environmental applications for satellite and airborne use. An airborne demonstration instrument was constructed, and the studies performed here reflect anticipated performance of the airborne demonstration device. We have added a sentence to clarify that.

Changes: Added sentence page 3 line 3 "The study presented here utilizes the specifications of the MURI airborne demonstration instrument"

2. Section 2.3 (Normalized Differential Methane Index). The method does not account for varying ground emissivity, the background vs gas temperature contrast, or the H2O and N2O column densities (are there strong absorption lines from these in the SB1 and SB2 bands?). These would affect the efficiency to detect CH4 (and for sure retrieval of column densities if this is a goal). If only detection, there could be false alarms from e.g. high H2O concentrations (which has lines overlapping the 7.7 micron CH4 band). Emissivities could also be different between different background materials.

AR2. Your comment does identify some of the limitations of our approach for which we are aware. However, because the approach utilizes a relative measurement the effect of surface emissivity and surface temperature changes will be less than with an absolute measurement. There are H2O absorption lines present, but there are considerably fewer and weaker features than the methane absorption lines in the same region. Band 6 also contains weak H2O absorption lines. The effect of H2O on masking detection using NDMI has not been fully characterized and could be the subject of a future investigation.

3a. Section 4.1. There are three relevant temperatures for this test: background, ambient, and plume. The efficacy to detect a plume would be very dependent on the background - plume temperature contrast, and if this contrast is 0 deg the plume could not be detected regardless of the sensitivity of a sensor as all the CH4 absorption lines C2 in the plume emits as much light as is removed (resulting in no absorption and no detectable difference in brightness temperature). In Figure 5, the 0 K curve (plume - ambient contrast is 0) would be a horizontal line if Tambient = Tbackground. It should thus be made more clear what temperature difference has been assumed between the ambient and background. Also, caption to Fig. 5 should explain that the curves are different contrast of ambient and plume temperature. In winter it could very well be emissions features, with the plume increasing the brightness temperature with e.g. a background of ice on a lake. The sentence (P11, first row) "The results here indicate that a plume with a temperature difference as high as +10 K to ambient temperature (which is not given as the ambient temperature is not given). This should be made more clear.

AR3a. Of the three relevant temperatures, Figure 5 describes the difference between plume temperature and ambient atmospheric temperature. In this scenario ambient atmospheric temperature does not equal background surface temperature and this has been clarified in Table 2. These were framed as plume to ambient atmospheric temperature differences as this is how our models are defined and for consistency throughout the paper. The background surface temperature in all cases is higher than plume temperature and therefore absorption is to be expected and this has been clarified. Supplemental material includes Figure 5 in terms of the temperature difference between the plume and the background surface.

Changes: Table 2, now table 3 includes ambient atmospheric temperature.

Page 11 line 12-13 added "This is a range of -27 to -7 K to the background surface."

Line 17 - 18 "which is consistent with the knowledge that the background surface temperature is 17 K higher than the ambient atmospheric temperature"

Table 4: contains ambient atmospheric temperature and surface temperature

Page 14 Line 13: "ambient atmospheric temperature" now reads "background surface temperature"

3b. Similarly, in the conclusions (P12) it is stated "The single band investigation confirmed that methane plumes with large concentrations and temperature differences compared to ambient atmospheric conditions lead to detectable contrasts". I agree with sufficiently large concentrations, but the important temperature difference is not the ambient and plume temperatures, it is the background and plume temperatures. This should be made more clear/rewritten. One could easily have the case of a very large plume-ambient temperature difference (say 10 K) but also a 10 K background ambient temperature difference, leading to no absorption lines and no difference in brightness temperature.

AR3b. This comment is greatly appreciated. Our use of ambient to plume temperature as the point of comparison was based off how our models are defined and also allowed us to maintain consistency. However, clarifying our results by discussing the background/plume contrast is included in our revision. Additionally, a supplemental figure showing our results in terms of plume/background temperature difference has been included.

Changes: See changes from AR3a.

Technical corrections: - Wrong table number. Page 9, 4.1. states "Table 3 contains...", this should be Table 2? (there is no Table 3)

Changes: Additional table added, original Table 2 is now Table 3, references table reflect proper label

- Abstract: 7.68 um -> 7.68  $\mu$ m. Also in other parts of the text (e.g. Table 1) using u instead of  $\mu$  - Table 1: Write  $\mu$ m in the headers instead of every row

Changes: um changed to  $\mu m$  throughout the text

Introduction: Use CH4 instead of methane after first having introduced "methane (CH4)". This is also the case for many other parts of the text.

Changes: "methane" revised to "CH4" throughout revision

- P5L17. "from pair" - > "from pairs"

Changes: "from pair" revised to "from a pair"

AR Technical corrections: Thank you very much for addressing these technical errors. These have been addressed in our revision.

**Referee #2**

Referee #2 Comments Response: On behalf of myself and my coauthor I would like to thank you for taking the time to review and provide feedback on our paper. Your feedback was very helpful and much appreciated. We know that your feedback will help make our revision a better discussion of our work.

General Comments The manuscript (Webber and Kerekes 2020) compares the performance of three different analytical methods for detecting methane in remote sensing imagery taken using an uncooled multispectral infrared (IR) radiometer. Given the prohibitive cryogenic requirements of traditional thermal IR imagers, an uncooled instrument would lower barriers to deploying imagers for atmospheric methane detection. This paper provides a useful evaluation of this system for methane detection; however, the description of the methodology, and the discussion and conclusions require more development. In particular, more quantitative details about assumptions made and model input used should be included, and reasons for the values chosen should be explained

Author Response(AR): Thank you for your comments pointing out the potential benefits of an uncooled instrument and recognizing the utility of our studies. We have added details to the descriptions of the methodology and enhanced our discussions to clarify points raised by the reviewers.

Specific Comments Page 2 I. 9-10: The phrasing that HyTES has been used to develop an algorithm that can predict methane concentration from thermal imagery is somewhat vague and therefore confusing. It would be more helpful to identify the improvements in the HyTES retrieval algorithm in Kuai et al. (2016) that are most relevant to the research described in this paper

ARI: 9-10 Some of the data used to inform the models we used in this study is from Kuai et al. We reference this study here to give context of what thermal instruments have been used for in the past as attempting to retrieve methane concentration is out of the scope of this study.

I. 19: Given that sensors that operate in various regions of the IR spectrum are discussed, it would be helpful to briefly clarify why traditional thermal IR sensors require cooling and the advantage of thermal IR over shortwave infrared (SWIR) sensors, which also measure methane but do not have the same cooling requirements.

ARI:19 A brief discussion of the advantages of TIR over SWIR was added and a statement of why traditional thermal IR sensors require cooling was added to this section

Changes: Page 2 Line 6 -10 and 21 – 26 now include discussion of advantages and disadvantages of TIR

I. 21: What defines a "satisfactory performance"? What is the level of sensitivity, precision, accuracy, or another relevant metric needed for methane detection applications of MURI?

ARI:21 No specific quantitative metrics for environmental applications were defined for the MURI project. Rather, satisfactory performance was defined more qualitatively as the system demonstrating useful performance in environmental applications. The project was conceived primarily as a technology development effort. This has been clarified in our revision.

Changes: page 2 line 34-35 previously read: "while maintaining a satisfactory performance in the thermal region of the infrared"

Now read: " while demonstrating the ability to record remote imaging data that is valuable for environmental applications."

I. 23: What is the difference between the airborne and satellite system? Are they using the same FPA?

ARI:23 The airborne and the satellite system design utilize the same focal plane array and similar optics with an effective focal length of 120 mm and an fnumber of 1. The work presented here is focused on the airborne system and we have clarified that in our revision.

Changes: Page 3 line 3 added sentence "The study presented here utilizes the specifications of the MURI airborne demonstration instrument."

Page 3: l. 13: What assumptions were made about environmental conditions, particularly the concentrations of interfering molecules such as water vapor?

ARI:13 This has been clarified in the Data Set Creation section with environmental conditions specified in Tables 2, 3, and 4.

Changes: Added Table 2, page 7

Table 2. MODTRAN Parameter Settings for Validation of HyTES Simulated Radiances

| Model Input                             | Value                |
|-----------------------------------------|----------------------|
| Atmosphere                              | Midlatitude Summer   |
| Water Vapor Scaling Factor              | 0.07                 |
| CH 4 Scaling Factor          | 0.4                  |
| Collection Height                       | 4.572km (15000 feet) |
| On Plume Emitting Surface Temperature   | 311.5 K       |
| Off Plume Emitting Surface Temperature  | 305 K         |
| Plume Thickness                         | 10 m          |
| Surface Emissivity                      | LAMB_SANDY_LOAM      |
| Plume Base Altitude                     | 1 0 m         |
| Ambient Temperature at Plume Altitude   | 293.5 K       |
| Plume Concentration                     | 6 ppm         |
| Plume Temperature Difference to Ambient | -7 .K         |

**Table 3, Page 11**

Table 3. NEdT Single Pixel Study MODTRAN Simulation Chosen Values Parameter Settings

| Constant                            | Value                |
|-------------------------------------|----------------------|
| Atmosphere                          | Midlatitude Summer   |
| Water Vapor Scaling Factor          | 0.10                 |
| Collection Height                   | 4.572km (15000 feet) |
| Emitting Surface Temperature        | 328 K                |
| Plume Thickness                     | 20 m                 |
| Surface Emissivity                  | LAMB_SANDY_LOAM      |
| Plume Height Base Altitude          | 10 m                 |
| Ambient Temperature at Plume Height | 311 K         |

**Table 4, page 13**

Table 4. Matched Filter and NDMI Dataset MODTRAN Simulation Parameter Settings

| Constant                            | Value                |
|-------------------------------------|----------------------|
| Atmosphere                          | Midlatitude Summer   |
| Water Vapor Scaling Factor          | 0.10                 |
| Collection Height                   | 4.572km (15000 feet) |
| Emitting Surface Temperature        | 333 K         |
| Plume Thickness                     | 20 m          |
| Surface Emissivity                  | LAMB_SANDY_LOAM      |
| Plume Base Altitude                 | 1 0 m         |
| Ambient Temperature at Plume Height | 315.4 K       |

Page 4: I. 5-9: More details are needed for the methodology, particularly what assumptions were made in modelling the background and plume-present cases and why those assumptions were chosen. A discussion of the sensitivity of the model output to these assumptions should be included here if some a priori knowledge of the sensitivity factored into the choice of assumptions, and/or in the Results/Discussion section if relevant to determining the validity of the results.

ARI:5-9 Modeling of the background was done by choosing parameters which would match the model output to the HyTES scene. Additional details have been included in section 3 to address the model parameters and their derivations.

Changes: Page 6 line 24 previously read "...chosen to model the simulated data after" Now reads "...chosen to reference in order to produce a more realistic simulated model" Page 6-7 added sentence "Surface level air temperature was retrieved from Weather Underground (www.wunderground.com) and was set to 293.5 K. The concentration of the plume was determined by Kuai et al to be 6 ppm (Kuai et al 2016). A list of notable model inputs is recorded in Table 2." Page 7 Table 2 added, please see above

Page 7 line 11 "HyTES images recorded over Kern County, California"

I. 8: What is the magnitude of the increased concentration of methane? How does this compare to the Noise Equivalent Concentration Length (NECL) and/or minimum detectable column density of the sensor?

ARI: 8 For the single pixel study, NEdT is chosen as the metric of comparison for detection as we are changing plume temperature and concentration and NECL will change for plumes of different temperature. Additional methane plumes were varied from 1 to 50 ppm. The graph was created using the following concentration plumes: 1, 5, 10, 20, 30, 50 ppm. The step sizes were chosen to ensure fine sampling such that the shape of the Tplume – Tbackground curves were not affected by the concentration intervals.

I. 13-15: Since only a single band is allocated to the methane feature, what is the purpose of the other bands? Section 2.3 demonstrates that the other bands can help constrain the methane retrieval, but if they have additional functions, those functions should be listed (in this paragraph, in the general description of the instrument, or in Table 1).

ARI:13-15: Additional information for each band has been added to Table 1 in order to describe their functions.

| Band #     | Center Wavelength (µm) | Band Width (µm)      | Application                                   | Predicted NEdT (K)  |
|------------|------------------------|----------------------|-----------------------------------------------|---------------------|
| B 1 | 7.68 <del>um-</del>    | 0.10 <del>um</del> - | $CH_4$                                        | 0.256 <del>K</del>  |
| B2         | 8.55 <del>um-</del>    | 0.35 <del>um</del> - | SO2, cloud/volcanic ash                       | 0.076 <del>K</del>  |
| B3         | 9.07 <del>um-</del>    | 0.36 <del>um</del> - | Minerals.SO2                                  | 0.078 <del>K</del>  |
| B4         | 10.05 <del>um-</del>   | 0.54 <del>um</del> - | Surface Temp. Retrieval, Vegetation, minerals | 0.059 <del>k</del>  |
| B5         | 10.90 <del>um</del>    | 0.59 <del>um</del> - | Surface Temp. Retrieval                       | 0.061 <del>K</del>  |
| B6         | 12.05 um-              | 1.01 um-             | Surface Temp. Retrieval                       | 0.036 <del>K.</del> |

**Changes: New column added to Table 1**

Table 1. MURI Band Allocations and Predicted Noise Equivalent delta Temperature

I. 21: Units associated with each of the variables would be helpful to conceptualize the relationships in Equation 3 and clarify what is meant by "signal", which can refer to multiple aspects of the data stream

ARI: 21 The signal is defined here as the spectral signature of a methane plume, which here is defined as absorption intensity, retrieved from hitran dataset. A clarifying statement as to what the signal is has been added. This assumed linear relationship is written in terms of sensor reaching radiance, so r and c are in terms of W/m2sr um, b is a absorption intensity, and  $\alpha$  is a scalar proportional to the plume strength.

Changes: Page 5 lines 1-2 "...in this case a methane plume absorption or emission signal..."

Page 5: I. 9-11: Is the threshold applied to the CMFI value, or some statistics associated with it, such as a confidence interval or t-stat? Also, please provide a short explanation of how the ROC curve is used to assess the effectiveness of the method.

ARI: 9-11 The threshold is applied to the CMFI value. The ROC curves describes the hit and false alarm rate at each concentration, indicating where in the concentration space the system can reliably differentiate between on and off plume pixels. Another method of assessing ROC curves is to calculate the area under the curves, which we have provided in the supplementary materials. Section 4.2 includes a discussion of how the ROC curves are used to assess the effectiveness of the method, and the section has been updated in response to your suggestion.

Change: Page 12 Line 15-16 "A straight line with a slope of 1 indicates that the detection scheme is performing as well as chance. Otherwise, a high hit rate and low false alarm rate indicate a reliably detectable scenario."

I. 23-27: This explanation is somewhat confusing. Is this paragraph describing whether the methane feature is giving an absorption versus emission signal in the detection? The way that NDMI is described, it seems like it would be possible to have negative values that can be indicative of a methane plume, and if no plume exists, the NDMI would be zero. If so, it seems that a higher absolute value of the NDMI would indicate C3 higher methane. Please clarify.

ARI: 23-27 Thank you for this comment, we have added to the paragraph to clarify. This paragraph is meant to describe how the NDMI will be different for a plume that is hotter than the surface and a plume that is cooler than the surface. If a plume is hotter than the surface, the resulting at sensor radiance for the methane feature band (SB1) will be higher than if there was no plume present. This would mean that, given a scenario in which all other variables remain the same, NDMI would be a lower value as the difference between SB2 and SB1 will be lower and the sum of SB2 and SB1 will be higher. If the plume is cooler than the background surface temperature, the plume will absorb more energy than it emits and the at sensor radiance for the methane feature band (SB1) will be lower than if there was not a plume present. NDMI would then be a higher number as the difference between SB2 and SB1 will be lower than the plume not present case. The NDMI is only 0 if SB1 and SB2 are equal and is not necessarily an indication of a plume presence or absence as SB1 and SB2 are defined as different spectral channels. It is possible to have a negative NDMI given a high enough temperature difference between the plume and the background surface. The NDMI is a relative measurement, and therefore can only be determined by comparing NDMI calculations across an image.

Changes: Page 6 line 4-6 now reads "Where the SB2 and SB1 bands are the radiance values recorded by two different spectral bands from the same instrument which cover the same spatial pixel area, one that includes a  $CH_4$  feature (SB1) and one that does not (SB2).

Page 6 I. 4: Please specify what band 2 has a comparatively higher transmission of: the atmosphere, instrument filter, etc.

ARI: 4 We have clarified that we mean the atmosphere has a higher transmission for band 2 than band 1.

Changes: page 6 line 15 "comparatively higher transmission" changed to "comparatively higher atmospheric transmission"

I. 17: It's unclear what is meant by "on and off plume spectra". Are these the spectra for a single background pixel and a different pixel that has a methane detection? Also, what are the assumptions that were made for the MODTRAN simulated recreation of the data? If these are the same assumptions used in Table 2, please refer to that table in this paragraph

ARI: 17 The on and off plume spectra describe two pixels taken from the HyTES imagery shown in Figure 1. The on plume pixel refers to a pixel identified by the HyTeS dataset to contain a methane plume, and the off plume pixel refers to a pixel identified by the HyTES data set to not include an enhanced level of atmospheric methane. Some assumptions are shared between this simulation and the original Table 2, a new table has been included to specify the chosen values.

Changes: Table 2 added, please see above.

l. 18: Based on the radiance values in Figure 3, the RMSE for the methane plume case is about 2.5% – How does this contribute to the uncertainty in the methane column density amount?

ARI: 18 Our single spectral band approach does not allow for quantification of methane, making column density uncertainties outside the scope of this study.

I. 19: How is "reasonable" defined? High confidence? If so, what is the threshold?

ARI:19 The language in our manuscript was unintentionally vague. We meant to say this recreation gave us insight into scenes where enhanced methane has been detected before and the confidence to utilize MODTRAN to create controllable simulations that resemble real data that has been used to detect methane previously. The language has been changed to better reflect the intended meaning.

Changes: Page 7 line 4-6 previously read, "Recreation of this data provided insight into reasonable scenes where methane could be detected and informed the model used in this study." Now reads, "Recreation of this data provided confidence that realistic scenes could be reproduced in MODTRAN6 and helped inform the input parameters for the other simulated datasets."

Page 8 I. 11: Is "only small amounts of CO2" referring to the ambient concentration input into MODTRAN? Please provide the actual value used and why it was chosen. If these results are not sensitive to the assumed concentration of CO2 please state that; if the chosen concentration of CO2 impacts the results, however, provide justification for the value chosen (e.g. regional average concentrations taken from in situ or satellite measurements).

ARI:11 As stated on line 13-14 (page 10 line 5 in the revision and page 9 line 14 in the recorded changes document) the effect of CO2 absorption is eliminated by our data set creation approach. The results are not sensitive to the assumed concentration of CO2 in this simulation. The chosen value is a small amount of CO2 that must be included to allow the MODTRAN simulation to run properly.

I. 12: What is the level of enhancement in the "enhanced concentration plume"? Please be quantitative.

ARI: 12 The enhanced concentration plumes refer to the varying quantities used in the experiments, which are identified in Figures 6, 7, and 8. The enhanced quantities vary from 1 - 20 ppm and this has been clarified in our revision.

Changes: page 9 line 13, "..ranging from 1-20 ppm above ambient methane."

Page 10 Table 2: Change "Plume Height" to "Plume Altitude", as the former could be confused with "Plume Thickness". Also, please add a row with the assumed ambient temperature

ARTable 2: Thank you, your suggestion has been added to Table 3 (formerly table 2).

Changes: Table 3 Height changed to Base Altitude, please see above

Section 4.1: Please provide rationale for the model inputs listed in Table 2 in this section. For instance, was the plume thickness derived from data, a model, or experiments? Was the ambient temperature measured locally, or taken from a model, and if so, which one? Assumptions that don't have a significant impact on results can be stated as such; however, justification should be provided for assumptions that alter the results and especially conclusions of this paper.

AR\_Section 4.1: Model assumptions were derived by adjusting MODTRAN inputs to produce simulated radiances that matched the HyTES data. Ambient temperature was recorded by a local weather station and retrieved from Wunderground. Additional rationale has been included to inform their reason for inclusion in the study in our revision.

Changes: page 11 table 3, ambient atmospheric temperature added, see above See response to Page 4: I. 5-9:

page 11 line 5, "Ambient atmospheric temperature was estimated from Weather Underground which was recorded by the Meadows Field Station in Bakersfield, California on July 8th, 2014 at 11:54 am." Page 12 line 7 – 9 " Ambient atmospheric temperature was retrieved from Weather Underground which was the daily high temperature recorded by the Meadows Field Station Bakersfield, California on July 8th, 2014. Surface temperature was determined by matching a blackbody to a selection of random pixels from the HyTES imager."

Page 11 I. 1-6: It is unclear why the sensitivity described was framed in terms of the temperature gradient between the plume and the ambient air, as it is the temperature gradient between the surface and the plume that drives the sensitivity in hyperspectral imagery. There might be a reason to frame the conclusions in the terms used, but it is difficult to evaluate those conclusions without knowing what ambient air temperature was chosen. The minimum detectable concentration of methane is lower when the thermal contrast between the plume and the surface is high; for example, a very hot plume should be more detectable over a low-temperature surface. Thus, the assertions made in this paragraph would not apply in all cases and would depend on the relative ambient, surface, and plume temperatures. Since the paper is evaluating the performance of new instrumentation, characterizing which conditions the conclusions hold for would be helpful in evaluating the applicability of these techniques for conditions that deviate from those chosen for this study.

ARI: 1-6. Results were framed as plume to ambient atmospheric temperature differences as this is how our models are defined. We have included the ambient temperature in our revision and have describe our results with context of surface and plume temperature differential. Additionally, a supplemental figure has been added to display the results in terms of the plume/background temperature difference.

Changes: Ambient temperature added to Tables, see above

Page 11 line 12-13, "This is a range of -27 to -7 K to the background surface." Page 11 line 17 – 18, "... which is consistent with knowledge that the background surface temperature is 17 K higher than the ambient atmospheric temperature." Page 12 line 4-5 "... or approximately -27.6 K from background surface temperature"

I. 12-14: When determining the false positive rate, what is used as truth? Is there ground-truth, or is the HyTES detection mask considered truth? Also, what is the region of interest threshold chosen? That is, does the algorithm require a certain number of contiguous pixels with methane detection before the plume is accepted?

ARI:12-14 As described in Section 3, MURI images were simulated using a HyTES image containing no enhanced concentration methane plumes. One image was created to be the background image, and a set of images were created simulating enhanced concentration methane plumes. We therefore have two separate images, one with methane and one without. This makes for an easy identification of hits, false alarms, correct rejections, and misses. Section 3 has been updated to clarify how the truth map is defined.

Changes: page 10 line 1-7 "The final dataset consists of five images derived from the scene depicted in Figure 4. The first image has no enhanced levels of methane present, the rest of the images have only enhanced levels of methane present across the entire image. Each image has a plume of constant concentration ranging from 1-20 ppm and plume temperature difference of -10 K from ambient atmospheric temperature. This also provides a simple truth map, as a perfect accuracy method would indicate the background image as having no methane present pixels (0 false alarms) and the plume present images as having every pixel be indicated as plume present (hit rate of 1)."

Technical Suggests: Page 1 l. 16-17:

A citation is missing after "While the concentration of methane is lower than that of CO2, the world has seen a rise in methane emissions since 2007, primarily from anthropogenic sources."

Changes: citation added

Page 2 I. 3: "Thrope" should be changed to "Thorpe".

Changes "Thrope" changed to "Thorpe

I. 5: It would be useful to specify that HyTES is a longwave infrared (LWIR) imager.

I. 18: Since the abbreviation for methane, CH4, is used earlier in the paper, it should be continue to be used consistently. This applies to the remainder of the paper, as well.

Changes: Methane revised to CH4 throughout

I. 11: A comma is missing after "(GOSAT)".

Changes: comma added after "(GOSAT)".

I. 17: A comma is missing after infrared".

Changes: comma added after infrared

I. 24: "FPA" should be changed to "focal place array (FPA)".

Changes: page 2 line 25, "FPA" specified as "focal plane array (FPA)"

I. 25: A comma is missing after "channels".

Changes: comma added after channels

I. 28: Both  $\mu$ m and um are used in this paragraph. One convention should be chosen for the entire paper.

Changes: um changed to  $\mu m$  throughout the text

Page 3 I. 5: Elsewhere in the paper, pixel is also used to describe both the physical pixel on the FPA (e.g. page 2) and the spatial pixel in the image (e.g. page 4). For clarity, change this instance and other references to the spectral pixel to "channel", which is used later in the paper.

Changes: page 3 line 17 "pixel" changed to "spectral band" page 4 line 25 added "spatial" Page 6 line 5 included word "spatial"

Page 4 I. 16: Add "spatial" between "N" and "pixel"

Changes page 4 line 25 added "spatial"

Page 5 l. 21-22: Specify whether SB2 or SB1 includes the methane feature.

Changes: page 6 line 5 now reads "... one that includes a CH4 feature (SB1) and one that does not (SB2).

Page 6 I. 11-12: Specify what "after" is referring to in the sentence "[...] a scenario in which a rogue emission source has been detected was chosen to model the simulated data after."

Changes: page 6 line 23 now reads "... a scenario in which rogue emission source has been detected was chosen to reference in order to produce a more realistic simulated model."

I. 15: Change "Prupulsion" to "Propulsion" (applies to other instances of this citation in the manuscript).

Changes: "Prupulsion" changed to "Propulsion"

Page 7 Figure 1: Specify units after "7.68". Also please add the ground sampling distance (GSD) of the

Changes: units added, GSD added

image. Page 8 l. 10: Remove the typo in "for at sensor radiance".

Page 9 I. 7: Change the reference to "Table 3" to "Table 2".

Changes: table references now correct

Page 10 l. 1-2: Citations are needed after "Modern estimates of ambient atmospheric methane concentration are at about 1.8 ppm, dangerous levels for 8 hours of daily exposure to methane for humans is 1000 ppm, while the lower explosive limit is around 50,000 ppm."

Changes: references added, paragraph updated, page 11 line 8-12.

I. 8: Remove "or" before "the methane feature band".

Changes: revision does not contain original error

Page 11 l. 15: An adjective is missing between "very" and "false".

Changes: "very" changed to "low"

Page 12 I. 14: Change "pixel" to "channel" or "band" if spectral pixel is what is meant.

AR Technical Suggestions: Thank you for the technical suggestions and for being so thorough. We are grateful and believe the additional technical suggestions you've provided have certainly improved the quality of the paper.

**Supplemental Materials**

Cody M Webber1 and John P Kerekes1

1Digital Imaging and Remote Sensing Laboratory, Rochester Institute of Technology, Rochester, NY 14623, USA **Correspondence:** Cody M Webber (cmw3698@rit.edu)

SM- 1. Recreation of Figure 5. Describes NEdT comparison study using temperature difference between the plume and the surface.

**ST- 1.** Area Under MURI Matched Filter ROC Curves (Figure 6)

| Concentration PPM # | Line Color | Area |
|---------------------|------------|------|
| 20                  | Black      | 0.97 |
| 15                  | Red        | 0.94 |
| 10                  | Blue       | 0.86 |
| 5                   | Pink       | 0.71 |
| 1                   | Cyan       | 0.55 |

| Concentration PPM # | Line Color | Area |
|---------------------|------------|------|
| 20                  | Black      | 0.86 |
| 15                  | Red        | 0.8  |
| 10                  | Blue       | 0.72 |
| 5                   | Pink       | 0.62 |
| 1                   | Cyan       | 0.52 |

ST- 2. Area Under MURI NDMI Band 1 and 2 ROC Curves (Figures 7)

ST- 3. Area Under MURI NDMI Band 1 and Band 6 ROC Curves (Figure 8)

[revised manuscript text omitted]

 (8)